# Factors beyond Body Mass Index Associated with Cardiometabolic Risk among Children with Severe Obesity

**DOI:** 10.3390/jcm13195701

**Published:** 2024-09-25

**Authors:** Ewa Kostrzeba, Mirosław Bik-Multanowski, Stephanie Brandt, Ewa Małecka-Tendera, Artur Mazur, Michael B. Ranke, Martin Wabitsch, Małgorzata Wójcik, Agnieszka Zachurzok, Anna Przestalska-Sowa, Elżbieta Petriczko

**Affiliations:** 1Department of Pediatrics, Endocrinology, Diabetology, Metabolic Disorders and Cardiology of Developmental Age, Pomeranian Medical University, 71-252 Szczecin, Poland; 2Department of Medical Genetics, Jagiellonian University Medical College, 30-663 Cracow, Poland; 3Institute of Human Genetics, University Hospital, LMU Munich, 80336 München, Germany; 4Division of Pediatric Endocrinology and Diabetes, Department of Pediatrics and Adolescent Medicine, Center for Rare Endocrine Diseases, 89075 Ulm, Germany; 5Department of Pediatrics and Pediatric Endocrinology, Medical University of Silesia, 40-752 Katowice, Poland; 6Department of Pediatrics, Pediatric Endocrinology and Diabetes, Institute of Medical Sciences, Medical College of Rzeszów University, 35-301 Rzeszów, Poland; 7Children’s Hospital in Tübingen, University of Tübingen, 72076 Tübingen, Germany; 8Department of Pediatric and Adolescent Endocrinology, Pediatric Institute, Jagiellonian University Medical College, 31-008 Cracow, Poland; 9Department of Pediatrics, Faculty of Medical Sciences, Medical University of Silesia in Zabrze, 41-800 Zabrze, Poland

**Keywords:** childhood obesity, severe obesity, cardiometabolic risk

## Abstract

**Background**: The increasing prevalence of severe obesity among children and adolescents poses a significant challenge for pediatricians and general practitioners. This study aimed to investigate the relationships between biochemical results, anthropometry, blood pressure measurements, and bioimpedance analysis (BIA)-derived parameters to identify potential cardiometabolic complications associated with severe obesity. **Methods**: This study included 347 children (162 boys, 185 girls) aged 0–19 years, meeting the criteria for severe obesity based on BMI thresholds for different age groups. The patients were recruited in four pediatric endocrinology centers in Poland (Zabrze, Cracow, Rzeszow, Szczecin). Each participant underwent anthropometric measurements, pubertal stage assessment, blood pressure measurement, biochemical and hormonal tests, and BIA. **Results**: BMI showed significant associations with fat mass percentage (FM%) and waist-to-height ratio (WHtR) but not waist-to-hip ratio (WHR). The relationship between BMI and FM% was stronger in girls and prepubertal children. The metabolic syndrome (MetS) Z-score showed a strong positive correlation with BMI in the pubertal children. A negative correlation between HDL and triglycerides was observed only in the boys. The prepubertal children exhibited more significant correlations, despite a smaller sample size and shorter duration of obesity. **Conclusions**: Considering multiple parameters beyond BMI alone provides a better understanding of cardiometabolic risks associated with severe obesity in children. MetS Z-score was not a reliable indicator of increased cardiometabolic risk in younger children. Early-onset severe obesity was associated with a higher risk of metabolic complications. Early intervention is crucial to mitigate metabolic complications in this population.

## 1. Introduction

Childhood and adolescent obesity has emerged as one of the foremost challenges in both the clinical and socio-economic domains during the 21st century. Currently, approximately one in four European children with obesity meets the criteria for severe obesity, resulting in nearly 380,000 children aged 6 to 9 years with severe obesity across 21 European countries [1]. This means that every general practitioner and pediatrician will increasingly encounter children with severe obesity in their medical practice. The prevalence of severe obesity among Polish children and adolescents remains unexplored to date. Approximately 40 thousand children aged 8–9 years suffered from severe obesity in 2015 in Italy [2]. According to data from 2018, the overall prevalence of severe obesity in 4–5 year-old children living in Wales was 3.1% [3]. In 2019, 34.8% of the Spanish population of pre-school children were classified as obese. Within this group, 3.5% of patients were categorized as having mild obesity, 1.2% were categorized as having severe obesity and 1.3% had morbid obesity [4]. The number of research studies performed on this study group is still very limited.

Obesity is directly connected with a comprehensive spectrum of over 200 complications, including disruptions in carbohydrate metabolism such as pre-diabetes and diabetes type 2, atherogenic dyslipidemia, hypertension, and Metabolic-Associated Fatty Liver Disease (MAFLD). Due to the increasing number of children and adults suffering from severe obesity and, consequently, its complications, there is a genuine necessity to find tools that facilitate the early detection of metabolic complications, enabling the implementation of a prophylactic approach or medical intervention [5]. Development of those strategies is particularly crucial for pediatricians and general practitioners, as these healthcare professionals continue to play a significant role in managing and coordinating the treatment of patients into adulthood. Overcoming obesity during pediatric age leads to decreases in a number of cardiometabolic biomarkers in young adults (normalization of cardiometabolic profile on average at 23 years of age) [6]. A study by Seo et al. showed that implementing a multidisciplinary lifestyle intervention program involving a 16-week exercise intervention targeted at children with moderate to severe obesity resulted in a decrease in the BMI Z-score by about 0.05 (*p* = 0.02). The exercise group had lower body fat percentage and cardiometabolic risk markers, as well as higher lean body mass and leg muscle strength compared to the usual care group [7].

Current diagnostic and monitoring tools for severe obesity have several limitations, particularly in identifying metabolic complications. Traditional measures such as Body Mass Index (BMI) and waist circumference are commonly used but are limited in their ability to differentiate between fat mass and lean mass or to assess the distribution of fat, which is critical in evaluating metabolic risk. Additionally, these tools do not account for individual variations in metabolism, genetics, or lifestyle factors that may influence health outcomes. Advanced imaging techniques like MRI or CT scans provide more precise assessments of fat distribution but are costly, less accessible, and involve exposure to radiation in the case of CT. Blood tests, which are used to identify markers of metabolic complications, may fail to capture the full complexity of metabolic health, especially in the early stages of disease. Overall, these limitations highlight a need for more comprehensive, personalized, and accessible diagnostic approaches.

Unlike previous studies that often rely solely on BMI and waist circumference to assess obesity-related risks, our research incorporates a broader range of parameters, including biochemical markers, bioimpedance analysis (BIA)-derived data, and detailed anthropometric measurements. This holistic approach enables a more nuanced understanding of cardiometabolic complications in children with severe obesity. Additionally, by identifying specific age- and sex-related differences, as well as highlighting the importance of early detection and intervention, our findings could lead to better conceptualization of targeted and effective strategies for preventing metabolic complications in pediatric populations.

## 2. Materials and Methods

This study is a part of prospective multi-center clinical investigation conducted across four specialized medical centers in Poland (Szczecin, Cracow, Zabrze, Rzeszow) and describes the preliminary results of the project. The intended sample size consists of 500 patients within the age range of 0 to 19 years, all exhibiting severe obesity with an early onset, hyperphagia, and food-seeking behaviors. This project aims to establish a Polish database of children with severe obesity, characterize this cohort clinically and biochemically, and evaluate the prevalence of monogenic obesity among Polish children, as well as to identify new mutations in obesity-related genes [8].

The study group consisted of 347 children and adolescents (162 boys and 185 girls) meeting the following criteria. Inclusion criteria: 1. Age 0–19 years. 2. The presence of severe obesity, defined as a BMI > 24 kg/m^2^ in a child below the age of 2 years, a BMI > 30 kg/m^2^ in children aged 2–6 years, a BMI > 35 kg/m^2^ in children aged 6–14 years, and a BMI > 40 kg/m^2^ in children aged >14 years or those with documented severe obesity in the past (meeting the described BMI criteria during a hospitalization or visit to an outpatient clinic within the six months prior to inclusion in the study). 3. Hyperphagia and food-seeking behaviors. Each participant above the age of 8 completed the Three-Factor Eating Questionnaire (TFEQ). For children below the age of 8, their caregivers completed the Child Eating Behaviour Questionnaire (CEBQ). All caregivers, regardless of the child’s age, completed the Hyperphagia Questionnaire for Clinical Trials (HQ-CT). 4. Written informed consent of the patient’s parent/guardian and patient above the age of 13 years to participate in the study.

Exclusion criteria: 1. Lack of written informed consent from the patients’ parents/guardians or patients above the age of 13 years. 2. Secondary cause of obesity: previously diagnosed genetic syndrome coexisting with obesity, treatment with medicine with known effects on weight gain (glucocorticoids, valproic acid, risperidone, and others), Cushing’s syndrome, and other secondary causes of obesity.

One child aged 9 months was included in this study, having a BMI of 20.7 because of excessive hyperphagia and food-seeking behavior.

The patients were recruited in four medical centers of pediatric endocrinology involved in childhood obesity management (Zabrze, Cracow, Rzeszów, and Szczecin) from inpatient and outpatient departments from 1 July 2022 to 21 November 2023. Statistical analysis was performed on the whole cohort, with differentiation based on the gender and sexual maturity level.

The data were obtained during a single visit or hospitalization at the study center. Each patient enrolled in the study:-underwent a physical examination with anthropometric measurements (body weight was measured to the nearest 0.1 kg on a certified medical scale, body height was measured to the nearest 0.1 cm on Harpenden stadiometer, waist circumference was measured at the level of midpoint between the lowest rib and iliac crest, and hip circumference was measured at the level of the greatest convexity of the buttocks on the back and with cardboard applied tangentially to the greatest convexity of the abdomen on the front by measuring tape to the nearest 0.5 cm)-underwent pubertal stage classification using the Tanner scale-had their blood pressure measured (SBP—systolic blood pressure, DBP—diastolic blood pressure); blood pressure was measured using a calibrated automatic blood pressure monitor with a cuff size appropriate to the arm size while each participant was in a sitting position following a 15 min rest period before the examination-underwent blood uptake for biochemical and hormonal tests in fasting (ALT—alanine aminotransferase, AST—aspartate aminotransferase, glucose, insulin, TGD—triglycerides, HDL—high-density lipoprotein, LDL—low-density lipoprotein, total cholesterol); deviation from the blood collection procedure was allowed if the patient provided documentation of having undergone listed laboratory tests within the last 6 months before the examination-went through BIA; deviation from BIA was allowed for children unable to cooperate (too young to follow the instructions); BIAs were conducted using TANITA MC-580 M S MDD, TANITA MC-780MA-N, and TANITA MC-780 P MA devices to measure fat mass (FM, %) and fat-free mass (FFM, %).

BMI, WHR, WHtR, and homeostatic model assessment for insulin resistance (HOMA IR) were calculated using the following formulas:-BMI = weight (kg) ÷ height (m^2^)-WHR = waist circumference (cm) ÷ hip circumference (cm)-WHtR = waist circumference (cm) ÷ height (cm)-HOMA IR = (glucose (mg/dL) × insulin (µIU/mL)) ÷ 405-BMI Z-Score was calculated using the Pediatric Z-Score Calculator online [9] for children aged 2–19 years-MetS Z-score was calculated using the MetS Z-score Calculator online [10]

In order to state the statistically significant differences between data gathered, including girls vs. boys and prepubertal vs. pubertal patients, a nonparametric Mann–Whitney test was used. The above analyses were performed on two subgroups: all the study participants (*n* = 347) and patients who had entered puberty (*n* = 301). In order to determine the presence of significant correlations between the analyzed parameters, Spearman’s rank correlations were calculated, and correlograms were generated for five subgroups: whole study population (*n* = 347), girls (*n* = 185), boys (*n* = 162), prepubertal patients (*n* = 46), and pubertal patients (*n* = 301). Statistically significant results were defined for *p* < 0.05. All calculations and graphics were generated using the R programming and statistical environment (R Foundation for Statistical Computing, Vienna, Austria, version: 4.3.2).

## 3. Results

The study group consisted of 347 children (185 girls and 162 boys) with average age 13.4 years (13.7 for girls and 13.1 for boys, *p* = 0.034). A total of 301 children had entered puberty (expressed by Tanner II in at least one assessed variable), and 46 were assessed as Tanner I (prepubertal period) *p* < 0.001. The average age of the children who had entered puberty was equal to 14.4, while the average age of the children in the prepubertal period was equal to 7. The mean BMI of the whole study population was equal to 40.1. The BMI Z-score and MetS Z-score on average were both equal to 2.7. Both parameters were higher in the group of children in the prepubertal period (3.1 and 3.0, respectively). The mean value of WHR in the study population was equal to 0.9, while the average value of WHtR was equal to 0.7. The average HOMA IR was equal to 6.4 (6.7 for children in the pubertal period and 4.8 for children in the prepubertal period, indicating increasing insulin resistance with age and puberty).

An analysis of the biochemical and hormonal test results revealed that within the studied population, the children had experienced a wide range of glucose concentration from hypoglycemia, normoglycemia (majority, mean = 88.6 mg/dL), and impaired fasting glucose. The glucose concentrations were higher in the boys, and this difference was statistically significant (*p* = 0.027). For insulin, the average value calculated for the whole study population was equal to 29.0 µIU/mL, indicating hyperinsulinemia (cut-off equal to 24.9 µIU/mL was adopted from the laboratory standards). There was a statistically significant difference in insulin concentration in the children in the pubertal (30.1 µIU/mL) and prepubertal periods (21.8 µIU/mL) *p* = 0.001. Regarding the lipid profile, the average total cholesterol of the whole study population was equal to 163.5 mg/dL, HDL 41.8 mg/dL, LDL 97 mg/dL, and TGD 134.5 mg/dL. The triglyceride concentrations were higher in the boys, and this difference was statistically significant (*p* = 0.032). Liver enzyme levels were also elevated, especially in the boys (*p* < 0.001), with AST being particularly elevated in the prepubertal group of children (*p* < 0.001). The average blood pressure of the whole study population was equal to 135/80, and higher values were observed in the group of children who had entered puberty (136/81 vs. 125/76 in prepubertal period) and in the boys (137/80 vs. 133/81 in the girls). Regarding the BIA-derived parameters, the FM of the whole study population was equal to 46.8% and was higher in the girls (48.5% vs. 44.8% in boys, *p* < 0.001), while FFM was equal to 53.2% but higher in the boys (55.2% vs. 51.5% in girls, *p* < 0.001). Detailed data about the selected characteristics, derived parameters, results of biochemical parameters and hormonal tests, blood pressure measurements, and BIA-derived parameters of the whole study group according to gender and pubertal period are presented in Table 1.

Due to the significant number of children in the pubertal period, a statistical analysis of the results according to gender was also conducted within this group. The results demonstrated that the boys had higher MetS Z-scores, BMI Z-scores (2.7 vs. 2.5 in the girls in both parameters *p* < 0.001), and HOMA IR (7.2 vs. 6.3 *p* = 0.007). The boys also had higher glucose (91 mg/dL vs. 88 mg/dL *p* = 0.006), insulin (32 µIU/mL vs. 29 µIU/mL *p* = 0.016), TGD (150 mg/dL vs. 124 mg/dL *p* = 0.047), ALT (39 vs. 29 U/I *p* < 0.001), AST (29 vs. 25 U/I *p* < 0.001), SBP (139 mmHg vs. 134 mmHg *p* < 0.001), and FFM (56% vs. 51% *p* < 0.001) levels, while the girls had higher HDL (43 mg/dL vs. 40 mg/dL *p* = 0.020) and FM (49 vs. 44% *p* < 0.001) levels than the boys. Detailed data are provided in Table 2.

When examining the correlations between the gathered parameters in the whole study population (*n* = 347), the strongest positive correlations were observed between HOMA IR and insulin (r = 1.0), cholesterol and LDL levels (r = 0.9), and between AST and ALT (r = 0.8). BMI was strongly associated with FM% (r = 0.5), MetS Z-score (r = 0.4), and WHtR (r = 0.5). No correlation between BMI and WHR was observed. There was also a significant correlation between WHR and WHtR indicators (r = 0.6), SBP and DBP (r = 0.5), glucose and HOMA IR (r = 0.4), and between cholesterol and TGD (r = 0.4). The most robust negative relationship was observed between FFM% and BMI (r = −0.5) and WHtR (r = −0.4) and FM% (r = −1). Detailed results of the correlation analysis are presented in Figure 1.

Correlations were also examined within the group of girls. A statistical analysis revealed that the relationship between BMI and FM% (r = 0.7) in the group of girls was stronger than in the whole study population. Additionally, only in the group of girls exhibited relationships between ALT and WHtR (r = 0.3), FM% and MetS Z-score (r = 0.5), and between FM% and DBP (r = 0.3). In the group of boys, fewer variations in the correlations were observed. Both positive and negative correlations were largely compatible with the trends observed in the whole study population. However, there was a strong negative relationship between HDL and TGD (r = −0.3) that was not observed in any other analyzed group, and the MetS Z-score was more strongly correlated with the BMI (r = 0.6).

Correlations were also assessed based on the pubertal stage of the study population. A total of 46 children who took part in this study were classified as stage I in Tanner scale (13 girls, 33 boys). In this group, a statistical analysis revealed more very strong positive correlations between BMI and other parameters, including insulin (r = 0.6), HOMA IR (r = 0.5), and DBP (r = 0.4). The relationship between BMI and FM% was stronger than in the whole study population (r = 0.8). WHR was associated not only with WHtR (r = 0.6) but also with AST and ALT (r = 0.5 and r = 0.4). There was a stronger correlation between SBP and DBP (r = 0.7) than in the whole study population. Interestingly, the MetS Z-score correlated negatively with FM (r = −0.3), insulin (r = −0.4), HOMA IR (r = −0.5), and cholesterol (r = −0.6). There was also a robust association between insulin and HOMA IR and ALT (r = 0.5 and 0.4), DBP (r = 0.4 and 0.6), and FM% (r = 0.5 for both insulin and HOMA IR). A graphical representation of the analysis is presented in Figure 2.

The study population consisted of 301 children (172 girls and 129 boys) who had entered the pubertal period (classified as Tanner II or more). A graphical representation of the analysis performed on the group of children in the pubertal period is presented in Figure 3. In this group, BMI was correlated with insulin (r = 0.3) and SBP (r = 0.3). It is worth noting that there was a strong positive correlation between MetS Z-score and BMI (r = 0.7) and MetS Z-score and glucose (r = 0.4).

## 4. Discussion

The ongoing global epidemic of obesity in both children and adults has underscored the necessity for tools to identify risks associated with obesity. The simplest tools for assessing nutritional status and potential complications of obesity include anthropometric measures such as BMI, WHR, and WHtR.

Various global organizations, such as the World Health Organization (WHO) [11] and the International Obesity Task Force (IOTF) [12], have established standards for assessing the nutritional status of children using age- and gender-specific BMI percentile thresholds. It is concerning that over a period of 17 months, we successfully enrolled nearly 350 children meeting the criteria for severe obesity (with an average BMI of 40.1 and a BMI Z-score of 2.7) in four Polish medical centers. What is even more troubling is that in the group of the youngest children who had not yet entered puberty (average age of 7 years), the calculated mean BMI value was equal to 34.9 (BMI Z-score 3.1, meaning more than 3 standard deviations above the mean BMI for age and gender). A study estimating the number of children with severe obesity in Poland has not been conducted yet; however, based on our results, it is evident that the scale of the problem is significant.

WHR and WHtR ratios provide a straightforward and rapid method for assessing abdominal obesity, which is associated with a higher risk of metabolic disorders and cardiovascular diseases. Unlike BMI, which may not distinguish between fat and muscle mass, WHR and WHtR are less influenced by these variations. The norms for WHR and WHtR vary slightly depending on the source and adopted standards. According to the WHO, abdominal obesity is recognized when the Waist-to-Hip Ratio (WHR) exceeds 0.85 in women and 0.9 in men [13]. Based on a recently published meta-analysis by Eslami et al., the optimal cut-off value of WHtR to predict central obesity in children and adolescents is 0.49 for both genders [14]. The average values of both WHR (0.9 for both genders) and WHtR (0.7 for both genders) in our study population exceeded the mentioned cut-off points, indicating the presence of abdominal obesity. It is interesting that no statistically significant correlation between BMI and WHR in our study population was found, while WHtR correlated very strongly with BMI and FM%, qualifying WHtR as a superior indicator of fat distribution. WHtR takes into account height, which helps standardize the measurement and provides a better reflection of how fat is distributed relative to a person’s overall stature [14]. According to the literature, WHtR exhibits a superior ability to forecast health risks associated with central obesity, including type 2 diabetes, hypertension, and cardiovascular disease, particularly in children and adolescents aged five and older [15].

In addition to classical anthropometric methods used to assess nutritional status and potential metabolic complications, our analysis also employed the MetS Z-score parameter and FM% obtained using bioimpedance method. Currently, there are no standardized diagnostic criteria that are consistently used to define metabolic syndrome in childhood and adolescence [16]. MetS is a constellation of cardiovascular risk factors linked to insulin resistance, believed to result from underlying processes involving dysfunction of adipocytes, systemic inflammation, and oxidative stress [17]. The MetS z-score in our analysis was derived from a confirmatory factor analysis that examines how the various components of MetS (obesity (BMI or WHR), blood pressure, TGD, HDL cholesterol, and fasting glucose) are correlated with one another [12]. Its clinical significance in the predicting future risk of developing cardiovascular disease and type 2 diabetes in children and adolescents remains a controversial issue [17]. Interpretation of the results is challenging, especially in the group of youngest children, as the MetS Z-score was developed in a group of US adults [10]. A MetS Z-score equal to 0 indicates an average degree of metabolic syndrome, and scores above 0 are associated with a greater risk of future disease, particularly scores above 1 (which is higher than 84.1% of US adults) or 2 (which is higher than 97.7% of US adults) [12]. The average MetS Z-score calculated for our population was very high—equal 2.7—indicating an increased risk of cardiovascular disease and diabetes type 2. We found strong positive correlations between the MetS Z-score and BMI and WHtR in the children who had entered puberty. Additionally, the MetS Z-score in girls was strongly correlated with FM%, which could result from the physiologically greater amount of body fat percentage in women [18]. On the contrary, the analysis of the prepubertal children’s data revealed strong negative correlations of MetS Z-score with glucose, insulin, HOMA-IR, and FM%. This confirms that application of this parameter in the group of youngest children is limited, as this indicator was originally developed in the group of adults.

Bioelectrical impedance analysis (BIA) has gained popularity due to its ease of use and non-invasiveness. It allows for rapid and efficient assessment of body composition by measuring the resistance of electrical currents as they pass through body tissues [19]. High body fat is linked to increased cardiometabolic risk and hypertension. BIA is also suggested as a potential predictor for pediatric MAFLD [20] and metabolic syndrome [21]. It has been proven to be applicable in the population of children with severe obesity [16,17]. A study performed on a group of 1998 people aged 16 to 91 years revealed increased cardiometabolic risk when body fat exceeded 25.9% for men and 37.1% for women [16]. Williams et al. provided a cut-off for total percentage of body fat equal to 30% for females and 25% for males as values significantly associated with cardiovascular risk factors in children and adolescents [22]. The average value of body fat in our study population was 48.5% for the girls and 44.8% for the boys (on the contrary, the boys experienced higher FFM values, equal to 55.2% vs. 51.5% for the girls). According to Bojanic et al. [23], in boys between the ages of seven and ten, there is a gradual increase in the average body fat percentage. However, after the age of 10, there is a consistent decline in body fat percentage until reaching 14 years of age. For girls, they observed a steady rise in average body fat percentage from the age of 6 and continuing until 14 [24]. These data correspond with our findings, as the average age of our study population was equal to 13.4. It is concerning that the total percentage of body fat in children in the prepubertal period was also very high, equal to 48.2%. The other BIA-derived parameter was FFM (which includes muscle mass). Children with obesity typically have a higher FFM compared to children with an average body weight, which can be attributed to the need for enhanced muscle mass to support excess weight and maintain mobility. Obesity often leads to inflammation and fluid retention, which can result in edema, impacting BIA measurements as well. Moreover, during periods of rapid growth, such as childhood and adolescence, children naturally experience an increase in FFM as part of their normal growth and development process [19]. It is crucial to recognize that while FFM may be higher, this does not necessarily equate to being healthy or desirable.

The average results of insulin, TGD, liver enzymes, and blood pressure exceeded the cut-off values. One particularly concerning fact is that the children in the prepubertal period achieved almost equally high test results compared to the children in the pubertal period. The maintenance of fasting glucose levels within the normal range is achieved through a compensatory mechanism involving hyperinsulinemia, as indicated by the HOMA-IR values. HOMA-IR provides a relatively straightforward approach to evaluating insulin resistance. Currently, there is no established threshold value indicating pathology for the HOMA-IR index in pediatric patients. Shashaj et al. suggested that values surpassing 1.68 in individuals with an average body weight suggest a “non-physiological state” (3.42 in the group of children and adolescents) and may elevate the risk of cardiovascular diseases [25]. The 2012 OSCA guidelines proposed a HOMA-IR value > 4.5 as the cutoff point for defining insulin resistance in children with excess body weight [26]. The average HOMA IR in our studied population indicated the presence of insulin resistance (6.4 in the whole study population, 6.7 for children in the pubertal period, and 4.8 for children in the prepubertal period). It is noteworthy that physiological insulin resistance occurs during adolescence, marked by a 25–50% decrease in insulin sensitivity. Unfortunately, excess body weight often leads to an exacerbation of insulin resistance during adolescence, which is a significant concern, especially in the case of children diagnosed with severe obesity at a young age. According to a study by Reihner et al., T2DM in children and adolescents often occurs in the presence of a strong family history and may not be related to obesity severity [22]. In our analysis of the whole study population, HOMA-IR was significantly correlated only with glucose and insulin. It is interesting that in the group of youngest children, HOMA-IR was additionally strongly correlated with BMI (r = 0.5), ALT (r = 0.4), DBP (r = 0.6), and FM (r = 0.5), presumably due to a greater genetic predisposition for the development of metabolic complications associated with obesity [24]. Children in the prepubertal period exhibited a greater number of strong correlations, even despite a smaller sample size and potentially shorter duration of obesity. The wide range of glucose concentrations observed, from hypoglycemia to impaired fasting glucose, highlights the need for regular monitoring of glucose levels. Pediatricians should be vigilant in assessing and addressing glucose dysregulation, as early intervention can prevent the progression to more severe metabolic conditions. Given the elevated insulin levels and prevalence of hyperinsulinemia, clinicians should consider incorporating regular insulin screening into routine evaluations for children with severe obesity.

According to AAP guidelines, total cholesterol in children should not exceed 170 mg/dL, LDL 110 mg/dL, and TGD 75 mg/dL, while HDL should be greater than 40 mg/dL [27]. In our study, abnormalities in triglycerides compared to other lipids were distinctly noticeable (146.7 mg/dL for boys and 124 mg/dL for girls), serving as the primary marker of lipid disorders associated with an improper diet and high consumption of simple sugars. ALT and AST, especially ALT, are considered biochemical markers of liver damage. NASPGHAN recommendations suggest using ALT as a screening test for Metabolic-Associated Fatty Liver Disease (MAFLD), with values of 22 U/L for girls and 26 U/L for boys as the upper limits of normal [28]. In our study population, the average values of ALT for the girls were equal to 28.8 U/L and 38.8 U/L for the boys, indicating a very high risk of MAFLD development. The diagnostic criteria for MAFLD include the identification of liver steatosis in histopathological examinations, imaging studies or serological markers of lipid accumulation, and the presence of 1 out of 3 following criteria: overweight or obesity, type 2 diabetes, or at least two parameters confirming metabolic dysregulation [29]. Recent studies emphasize the need for a more precise definition of MAFLD in children, especially in the context of more effectively identifying children with an increased cardiovascular risk [30]. The norms for arterial blood pressure in children depend on their age, gender, and height. The average arterial blood pressure in the studied population was 135/80, meeting the criteria for stage I hypertension in adults according to the American College of Cardiology/American Heart Association (ACC/AHA) guidelines from 2017 [31]. This indicates that the majority of the examined children had blood pressure significantly exceeding the norm for their gender, age, and height. According to the American Heart Association, the overall prevalence of hypertension in children is 2% to 5% [32]. Children with hypertension have a significant likelihood of developing hypertension in adulthood and experiencing detectable damage to target organs, especially hypertrophy of the left ventricle and stiffening of the blood vessels [32].

Regardless of gender and maturation stage, strong correlations were observed between HOMA IR and insulin concentration, cholesterol and LDL levels, BMI and FM, BMI and WHtR, ALT and AST, and WHR and WHtR indicators, as well as between SBP and DBP. A strong negative correlation between HDL and TGD levels (r = −0.3) was observed only in the group of boys. This may suggest that obesity in the boys resulted from the consumption of more highly processed foods, while in the girls, excessive body weight could be associated with exorbitant calorie intake. Pediatricians should consider gender-specific risk factors and treatment strategies when managing severe obesity. For instance, the higher triglyceride and liver enzyme levels in boys suggest a need for targeted strategies to address these concerns, possibly through dietary modifications and lifestyle interventions. In our project, we investigated children’s dietary habits through surveys, and we will aim to address this topic in subsequent publications. Our findings indicate significant differences in metabolic parameters between prepubertal and pubertal children. Elevated blood pressure and insulin levels during puberty highlight the importance of monitoring these parameters closely during this developmental stage. Pediatricians should be aware that pubertal children with severe obesity may require more intensive management to address the increased risk of cardiovascular and metabolic complications. Early puberty may also necessitate a more proactive approach in managing metabolic syndrome, as indicated by the strong correlations between MetS Z-score and various metabolic markers. The examination of the prepubertal period population yielded particularly intriguing insights. The most likely primary contributors to the development of obesity in this age group are genetic factors, which predispose patients to the early development of metabolic complications. This was reflected in our results, as we observed a higher number of stronger correlations despite the smaller size of the group and potentially shorter duration of the metabolically adverse effects of higher body mass. Evaluation of the prevalence of monogenic forms of obesity in this cohort, with a special focus on leptin–proopiomelanocortin pathway abnormalities, will be investigated in the next stage of this study.

Childhood obesity increases the likelihood of persistent obesity in the long term, along with an elevated risk of significant complications and mortality in adulthood. Severe obesity during adolescence is linked to notably higher complication risks [24]. While treating childhood obesity is recommended and effective, intensive therapies like medically supervised meal replacement, pharmacologic treatment, and bariatric surgery come with risks and high costs. This study’s results emphasize the value of a multi-faceted approach to assessing severe obesity. Relying solely on BMI is insufficient, as it does not capture the full complexity of metabolic health. Pediatricians should adopt a comprehensive assessment approach, incorporating measurements of FM, FFM, and biochemical markers to better understand and manage severe obesity. The observed correlations between metabolic markers and obesity parameters, such as the strong relationship between BMI and FM%, suggest that guidelines should be updated to reflect a more nuanced understanding of severe obesity. Preventive strategies should focus on early identification of metabolic risks and personalized treatment plans that address specific metabolic abnormalities. For instance, incorporating regular screening for insulin resistance and dyslipidemia into routine care could improve early detection and management of potential complications. Our study highlights several areas for future research, including the need for longitudinal studies to track the long-term outcomes of early interventions and the development of gender-specific treatment protocols. Further research could also explore the effectiveness of different management strategies in reducing metabolic risks and improving overall health outcomes for children with severe obesity. In summary, the practical implications of our study suggest that pediatricians and general practitioners should adopt a more comprehensive, individualized approach to managing severe obesity in children. By integrating detailed metabolic assessments and considering gender and developmental stage differences, clinicians can enhance the effectiveness of their interventions and improve the overall health outcomes for this vulnerable population.

## 5. Conclusions

The results have shown that considering multiple parameters beyond BMI alone provides a better understanding of the cardiometabolic risks associated with severe obesity in children. MetS Z-score could be successfully applied in the group of children who had entered puberty, but it was not a reliable indicator of increased cardiometabolic risk in younger children. The earlier the onset of severe obesity, the greater the risk of metabolic complication development. Early intervention is crucial to mitigate metabolic complications in this population.

## Figures and Tables

**Figure 1 jcm-13-05701-f001:**
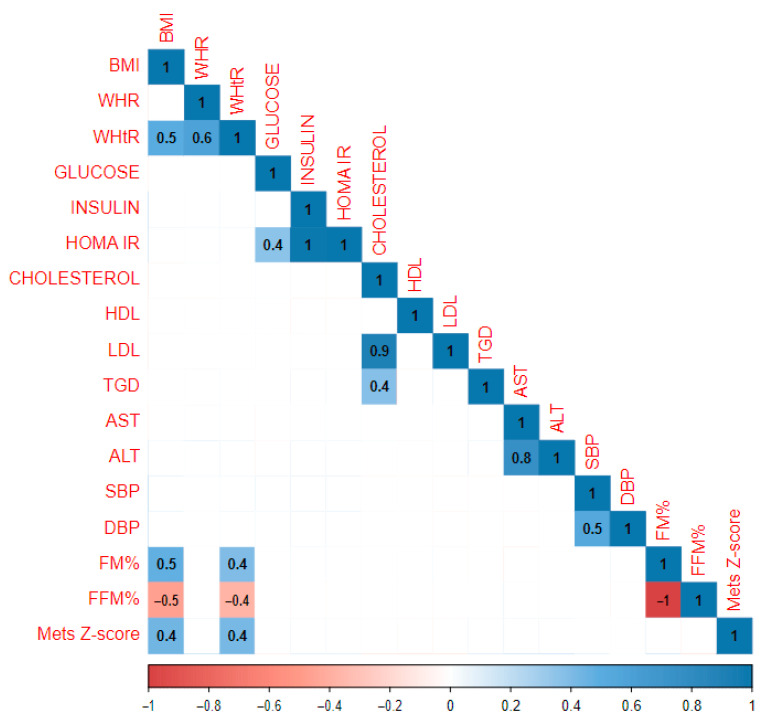
Results of the statistical analysis performed on the whole study population. Positive correlations between parameters are denoted in blue, with stronger correlations indicated by more intense blue coloring. Negative correlations are indicated in red, with more intense red coloring signifying stronger negative correlations between the observed parameters.

**Figure 2 jcm-13-05701-f002:**
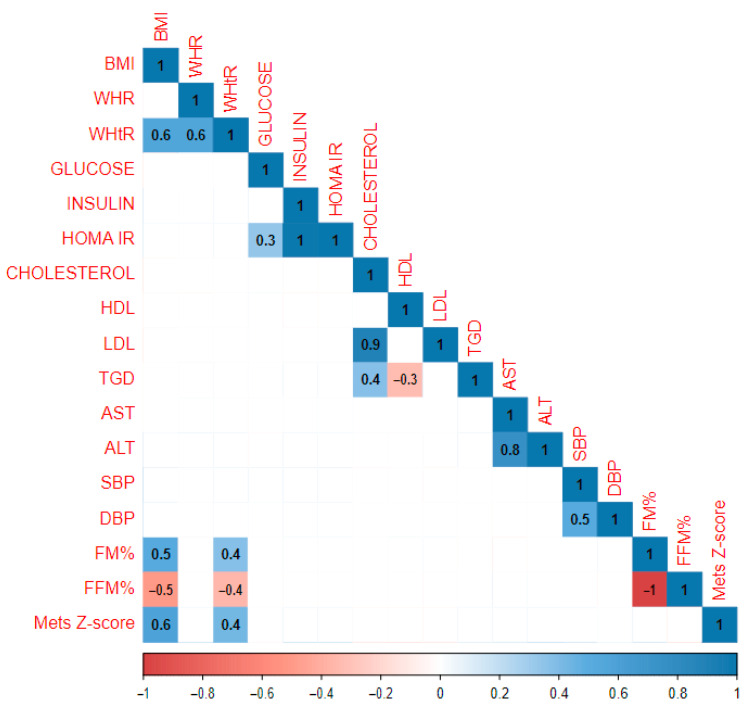
Results of the statistical analysis performed on the population in the prepubertal period.

**Figure 3 jcm-13-05701-f003:**
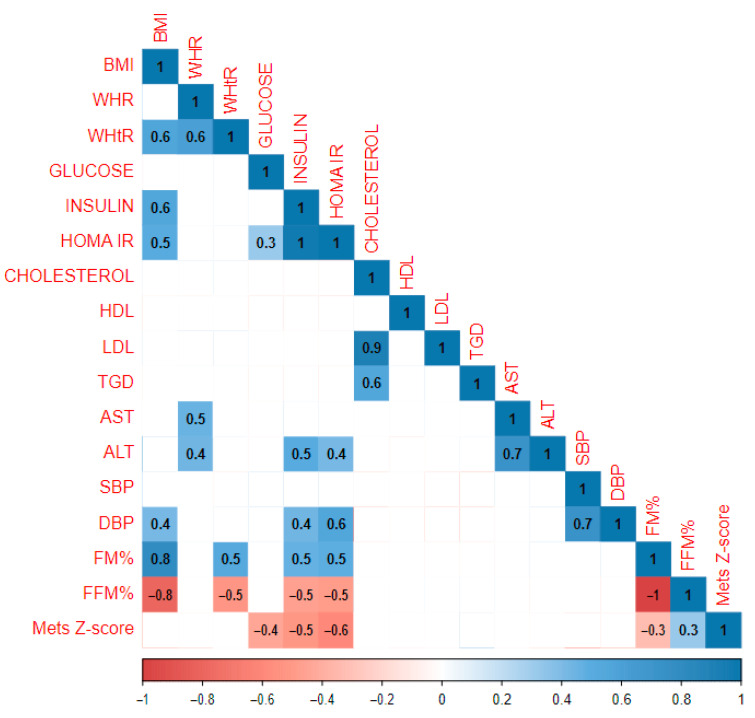
Results of the statistical analysis performed on the population in the pubertal period.

**Table 1 jcm-13-05701-t001:** Selected characteristics, derived parameters, results of biochemical and hormonal tests, blood pressure measurements, and bioimpedance analysis-derived parameters of the whole study population. ^1^ Range (Mean/SD), ^2^ Wilcoxon rank sum test. Statistically significant correlations of the characteristics with the gender and pubertal period are marked in red (*p* < 0.05).

Whole Study Population			Gender			Pubertal Period		
Characteristic	Available Data	Overall *n* = 347 ^1^	Girls *n* = 185 ^1^	Boys *n* = 162 ^1^	*p*-Value ^2^	Puberty *n* = 301 ^1^	Prepuberty *n* = 46 ^1^	*p*-Value ^2^
Age	347	0.8–18.9 (13.4/3.4)	0.8–18.9 (13.7/3.4)	0.8–18.6 (13.1/3.4)	0.034	7.8–18.9 (14.4/2.0)	0.8–14.3 (7.0/3.7)	<0.001
BMI	347	20.7–65.8 (40.1/5.9)	20.7–57.1 (40.1/5.7)	24.0–65.8 (40.1/6.1)	0.5	25.5–65.8 (40.9/5.3)	20.7–62.1 (34.9/6.9)	<0.001
BMI Z-score	340	1.7–5.5 (2.7/0.4)	1.7–4.9 (2.6/0.3)	2.1–5.5 (2.8/0.4)	<0.001	1.7–3.4 (2.6/0.2)	2.5–5.5 (3.1/0.7)	<0.001
Unknown		7	4	3		0	7	
Mets Z-score	310	1.7–5.5 (2.7/0.3)	1.7–3.2 (2.5/0.2)	2.1–5.5 (2.8/0.4)	<0.001	1.7–3.4 (2.6/0.2)	2.5–5.5 (3.0/0.6)	<0.001
Unknown		37	19	18		25	12	
WHR	340	0.5–4.5 (0.9/0.2)	0.7–4.5 (0.9/0.3)	0.5–1.2 (0.9/0.1)	<0.001	0.5–4.5 (0.9/0.2)	0.5–1.2 (1.0/0.1)	<0.001
Unknown		7	3	4		6	1	
WHtR	340	0.3–1.0 (0.7/0.1)	0.5–1.0 (0.7/0.1)	0.3–1.0 (0.7/0.1)	0.2	0.3–1.0 (0.7/0.1)	0.4–0.9 (0.7/0.1)	0.005
Unknown		7	3	4		6	1	
HOMA IR	319	0.5–19.1 (6.4/3.6)	1.2–19.1 (6.2/3.8)	0.5–17.3 (6.7/3.5)	0.057	1.2–19.1 (6.7/3.7)	0.5–15.3 (4.8/2.9)	0.002
Unknown		28	14	14		23	5	
Glucose (mg/dL)	338	63.0–123.3 (88.6/9.2)	64.0–122.4 (87.6/9.0)	63.0–123.3 (89.7/9.3)	0.027	64.0–123.3 (88.8/8.9)	63.0–122.4 (87.4/11.0)	0.2
Unknown		9	3	6		8	1	
Insulin (µIU/mL)	319	2.5–98.5 (29.0/15.6)	5.9–98.5 (28.2/16.0)	2.5–83.0 (30.0/15.0)	0.093	6.1–98.5 (30.1/15.7)	2.5–63.2 (21.8/12.3)	0.001
Unknown		28	14	14		23	5	
Cholesterol (mg/dL)	336	82.5–307.0 (163.5/32.0)	89.0–307.0 (163.7/31.1)	82.5–237.0 (163.2/33.0)	0.8	82.5–248.7 (162.8/31.1)	103.0–307.0 (168.0/37.6)	0.5
Unknown		11	6	5		8	3	
HDL (mg/dL)	335	25.0–64.0 (41.8/8.3)	26.0–64.0 (42.6/8.5)	25.0–59.2 (40.9/8.1)	0.089	25.0–64.0 (41.8/8.3)	25.3–58.0 (42.1/8.4)	0.7
Unknown		12	6	6		9	3	
LDL (mg/dL)	334	34.9–225.0 (97.0/27.0)	34.9–225.0 (97.6/26.8)	44.8–161.0 (96.4/27.3)	0.4	34.9–161.0 (96.5/25.9)	35.9–225.0 (100.3/33.9)	0.7
Unknown		13	8	5		10	3	
TGD (mg/dL)	332	10.5–597.0 (134.5/69.5)	10.5–361.4 (124.0/52.5)	40.0–597.0 (146.7/83.8)	0.032	10.5–597.0 (134.6/70.4)	46.7–335.0 (133.6/63.8)	>0.9
Unknown		15	6	9		12	3	
ALT (U/L)	341	7.5–194.0 (33.5/22.0)	7.5–104.0 (28.8/15.9)	10.9–194.0 (38.8/26.4)	<0.001	7.5–194.0 (33.0/21.8)	10.9–133.0 (36.6/23.2)	0.2
Unknown		6	3	3		6	0	
AST (U/L)	329	8.9–93.8 (27.5/11.6)	8.9–66.9 (25.1/9.9)	14.0–93.8 (30.1/12.7)	<0.001	8.9–93.8 (26.5/11.1)	16.0–84.0 (33.6/12.6)	<0.001
Unknown		18	11	7		16	2	
SBP (mmHG)	323	70.0–186.0 (134.9/15.3)	100.0–186.0 (133.1/15.4)	70.0–174.0 (137.0/15.0)	0.007	100.0–186.0 (136.2/14.7)	70.0–174.0 (125.0/16.3)	<0.001
Unknown		24	13	11		15	9	
DBP (mmHg)	323	40.0–118.0 (80.5/10.7)	56.0–110.0 (80.9/10.0)	40.0–118.0 (80.1/11.5)	0.7	40.0–118.0 (81.1/10.2)	40.0–116.0 (76.4/13.6)	0.023
Unknown		24	13	11		15	9	
Fat mass (%)	275	14.7–68.0 (46.8/7.1)	32.7–68.0 (48.5/6.1)	14.7–62.2 (44.8/7.7)	<0.001	14.7–68.0 (46.7/7.1)	25.2–61.5 (48.2/8.1)	0.2
Unknown		72	38	34		50	22	

**Table 2 jcm-13-05701-t002:** Selected characteristics, derived parameters, results of biochemical and hormonal tests, blood pressure measurements, and bioimpedance analysis parameters of the children who had entered puberty. ^1^ Range (Mean/SD), ^2^ Wilcoxon rank sum test. Statistically significant correlations of the characteristics with gender are marked in red (*p* < 0.05).

Children in Pubertal Period			Gender		
Characteristic	Available Data	Overall *n* = 301 ^1^	Girls *n* = 172 ^1^	Boys *n* = 129 ^1^	*p*-Value ^2^
Age	301	7.8–18.9 (14.4/2.0)	7.8–18.9 (14.4/2.2)	10.3–18.6 (14.4/1.8)	0.6
BMI	301	25.5–65.8 (40.9/5.3)	25.5–57.1 (40.8/5.1)	30.0–65.8 (41.0/5.6)	0.8
BMI Z-score	301	1.7–3.4 (2.6/0.2)	1.7–3.0 (2.5/0.2)	2.1–3.4 (2.7/0.2)	<0.001
Mets Z-score	276	1.7–3.4 (2.6/0.2)	1.7–3.0 (2.5/0.2)	2.1–3.4 (2.7/0.2)	<0.001
Unknown		25	13	12	
WHR	295	0.5–4.5 (0.9/0.2)	0.7–4.5 (0.9/0.3)	0.5–1.2 (0.9/0.1)	<0.001
Unknown		6	3	3	
WHtR	295	0.3–1.0 (0.7/0.1)	0.5–1.0 (0.7/0.1)	0.3–1.0 (0.7/0.1)	0.2
Unknown		6	3	3	
HOMA IR	278	1.2–19.1 (6.7/3.7)	1.2–19.1 (6.3/3.8)	2.1–17.3 (7.2/3.5)	0.007
Unknown		23	11	12	
Glucose (mg/dL)	293	64.0–123.3 (88.8/8.9)	64.0–117.0 (87.5/8.7)	72.7–123.3 (90.5/9.0)	0.006
Unknown		8	3	5	
Insulin (µIU/mL)	278	6.1–98.5 (30.1/15.7)	6.1–98.5 (28.7/16.1)	9.9–83.0 (32.0/15.0)	0.016
Unknown		23	11	12	
Cholesterol (mg/dL)	293	82.5–248.7 (162.8/31.1)	89.0–248.7 (163.4/29.6)	82.5–237.0 (162.1/33.0)	0.5
Unknown		8	5	3	
HDL (mg/dL)	292	25.0–64.0 (41.8/8.3)	26.0–64.0 (42.9/8.5)	25.0–59.2 (40.4/8.0)	0.020
Unknown		9	5	4	
LDL (mg/dL)	291	34.9–161.0 (96.5/25.9)	34.9–157.9 (97.2/25.0)	44.8–161.0 (95.7/27.0)	0.3
Unknown		10	7	3	
TGD (mg/dL)	288	32.9–597.0 (135.0/70.2)	32.9–361.4 (124.4/50.5)	40.0–597.0 (149.6/88.5)	0.047
Unknown		13	6	7	
ALT (U/L)	295	7.5–194.0 (33.0/21.8)	7.5–104.0 (28.9/16.4)	11.8–194.0 (38.5/26.5)	<0.001
Unknown		6	3	3	
AST (U/L)	285	8.9–93.8 (26.5/11.1)	8.9–66.9 (24.7/9.6)	14.0–93.8 (28.9/12.4)	<0.001
Unknown		16	10	6	
SBP (mmHG)	286	100.0–186.0 (136.2/14.7)	100.0–186.0 (133.8/15.3)	111.0–174.0 (139.4/13.4)	<0.001
Unknown		15	8	7	
DBP (mmHg)	286	40.0–118.0 (81.1/10.2)	56.0–110.0 (81.1/10.0)	40.0–118.0 (81.0/10.5)	0.9
Unknown		15	8	7	
Fat mass (%)	251	14.7–68.0 (46.7/7.1)	32.7–68.0 (48.7/6.2)	14.7–62.2 (44.1/7.3)	<0.001
Unknown		50	30	20	
Fat-free mass (%)	250	32.0–85.3 (53.4/7.1)	32.0–67.3 (51.3/6.2)	37.8–85.3 (56.0/7.3)	<0.001
Unknown		51	31	20	

## Data Availability

The raw data supporting the conclusions of this article will be made available by the authors on request.

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
