# Peer review of "Factors beyond Body Mass Index Associated with Cardiometabolic Risk among Children with Severe Obesity"

_jcm, 2024, doi:10.3390/jcm13195701_

Round 1

Reviewer 1 Report

Comments and Suggestions for Authors

The study by Kostrzeba et al. provides valuable insights into the cardiometabolic risks associated with severe obesity in children by examining a range of parameters beyond BMI, such as biochemical markers and bioimpedance analysis. The manuscript effectively highlights the importance of early intervention and comprehensive assessment in managing severe obesity, though further revisions could enhance its depth and applicability in clinical settings.

Provide recent statistics or trends on severe obesity in children globally, and compare them with the Polish context to highlight the severity of the issue.

Expand on evidences on how early intervention can prevent or mitigate long-term health issues associated with severe obesity.

Briefly discuss the limitations of current diagnostic and monitoring tools for severe obesity and their effectiveness in identifying metabolic complications in introduction.

Introduction will benefet by adding how your study’s approach or findings could differ from or improve upon existing research, especially in terms of diagnostic accuracy or preventive strategies.

Provide more details on how hyperphagia and food-seeking behaviors were assessed or quantified, as these are subjective measures.

Specify the criteria for determining “documented severe obesity in the past” for the inclusion of children older than 14 years

Mention the specific type of scales and stadiometers used for anthropometric measurements to ensure reproducibility

how missing data or deviations from the standard procedures were managed and whether any imputation methods were used.

The results are well summarized; however, I believe some of the text in the results section merely repeats the content of the tables and figures. Additionally, the results section is overwhelmed with seven tables and five graphs, which impacts the readability of the paper. Please consider merging the tables if possible, or making some of them supplementary to improve clarity.

Discussion should expand on the practical implications of your study for paediatricians and general practitioners, including how your findings could influence clinical practice or guidelines.

Author Response

Dear Editor and Dear Reviewers,

I am writing to express my sincere gratitude for the thorough review of our manuscript titled

"Considering multiple parameters beyond BMI alone provides a better understanding of cardiometabolic risks associated with severe obesity in children.” Your insightful comments and constructive suggestions have been invaluable in enhancing the quality of our work. In response to your feedback, we have made numerous revisions to address the concerns raised during the review process. All changes in the manuscript have been marked in red to facilitate tracking. Here is a summary of the key modifications made to the manuscript.

Reviewer 1:

Comment: Provide recent statistics or trends on severe obesity in children globally, and compare them with the Polish context to highlight the severity of the issue.

Response: Thank you for this valuable comment. We agree that statistics on severe obesity in Polish children would be extremely valuable. This type of research has not yet been conducted, however, it is one of the purposes of our project 'Prevalence of Monogenic Obesity among Polish Children and Adolescents with Severe Obesity - Polish-German Study.' In addition to data about the incidence of severe obesity in the European Union we have included statistics on the prevalence of severe obesity in Wales, Spain, and Italy to highlight the global scale of severe obesity in children (page 2, lines 56-60).

Comment: Expand on evidences on how early intervention can prevent or mitigate long-term health issues associated with severe obesity.

Response: That is a meaningful observation. We agree with your comment and, following your suggestion, we have cited two studies on page 2, lines 71-79, demonstrating how multidisciplinary programs targeting children with severe obesity lead to improvements in the cardiometabolic profile, emphasizing that early intervention can result in complete normalization of the cardiometabolic profile by the age of 23.

Comment: Briefly discuss the limitations of current diagnostic and monitoring tools for severe obesity and their effectiveness in identifying metabolic complications in introduction.

Response: Thank you for this noteworthy suggestion. On page 2, lines 80-91, instead of listing diagnostic and monitoring tools, we described the limitations of these methods, highlighting the need for a more holistic and personalized diagnostic approach. Children with severe obesity remain an under-studied population that requires tools capable of identifying their increased risk of cardiometabolic complications.

Comment: Introduction will benefit by adding how your study’s approach or findings could differ from or improve upon existing research, especially in terms of diagnostic accuracy or preventive strategies.

Response: We appreciate this insightful comment. On pages 2-3, lines 92-100, instead of stating the purpose of our study, we have described how our research differs from existing studies and how it can enhance medical care for pediatric patients with severe obesity.

Comment: Provide more details on how hyperphagia and food-seeking behaviors were assessed or quantified, as these are subjective measures.

Response: We are grateful for this constructive feedback. On page 3, lines 117-121 we added a description of applied quantification methods: each participant above the age of 8 completed the Three-Factor Eating Questionnaire (TFEQ) questionnaire, for children below the age of 8, their caregivers completed the Child Eating Behaviour Questionnaire (CEBQ). All caregivers, regardless of the child’s age, completed the Hyperphagia Questionnaire for Clinical Trials (HQ-CT). There was no cutoff point based on which patients were qualified or disqualified from the study, the most important factor for the inclusion was the patients' and caregivers' subjective experience of hyperphagia and food-seeking behaviors.

Comment: Specify the criteria for determining “documented severe obesity in the past” for the inclusion of children older than 14 years.

Response: Thank you for this helpful input. Documented severe obesity in the past refers to meeting the BMI criteria described in the methodology during a previous hospitalization or visit to an outpatient clinic within the six months prior to inclusion in the study. This information  was implemented in the manuscript on page 3, lines 114-116.

Comment: Mention the specific type of scales and stadiometers used for anthropometric measurements to ensure reproducibility how missing data or deviations from the standard procedures were managed and whether any imputation methods were used.

Response: That is a useful remark. To ensure reproducibility, body weight was measured to the nearest 0.1 kg using a certified medical scale, and body height was measured to the nearest 0.1 cm with a Harpenden stadiometer (page 3, lines 137-139). No imputation methods were used. In the tables presenting the results, we have included the number of unknown results.

Comment: The results are well summarized; however, I believe some of the text in the results section merely repeats the content of the tables and figures. Additionally, the results section is overwhelmed with seven tables and five graphs, which impacts the readability of the paper. Please consider merging the tables if possible, or making some of them supplementary to improve clarity.

Response: Thank you for this insightful feedback. Based on it, we have consolidated the data into two tables and revised the results section to improve clarity of the manuscript.

Comment: Discussion should expand on the practical implications of your study for paediatricians and general practitioners, including how your findings could influence clinical practice or guidelines.

Response: Thank you for this valuable comment. We expanded the Discussion section by implementation of practical implications for paediatricians and general practitioners resulting from our study (page 12, lines 407-409; page 13, lines 442-445; page 13, lines 447-455; page 14, lines 469-490).

Reviewer 2:

Comment: Title can be condensed and improved as 'Factors beyond body mass index associated with cardiacm risk among obese children'.

Response: In accordance with the suggestion, we have changed the manuscript title to 'Factors Beyond BMI (Body Mass Index) Associated with Cardiometabolic Risk Among Children with Severe Obesity'.

Comment 2: Authors have taken wide range of age groups (o-19 yrs). Will it be better if the children are analysed in varying age groups?. All body changes may not occur at the same time.

Response: This is a valid point. We fully agree that maturation changes do not occur at the same time in all children, and assessing the results within specific age groups would be valuable. On the other hand, our study group consisted of children with extreme obesity who met the restricted inclusion criteria for the study, therefore further division into subgroups would reduce the statistical reliability of the obtained results. Analyzing changes within specific age groups could be a good idea for future research if a larger cohort of children can be gathered.

Comment 3: There are multiple Tables and figures. Can the date be presented in two Tables and two figures?. One for physical and another for biochemical

Response: Thank you for this valuable comment, that allowed for the consolidation of data and the improvement of the manuscript's clarity. We have consolidated the data into two tables: the first presenting data for the whole study population, and the second presenting data for the children who have entered puberty. We also reduced the number of figures to three, presenting the correlations observed in the entire population, as well as those obtained in children during the pubertal and prepubertal periods.

Comment 4: Usually Titles and legends of figures are placed below the figures.

Response: Thank you for pointing this out. We have transferred titles and legends below the figures.

Thank you once again for your valuable contribution to the enhancement of our manuscript. I believe that your comments have not only improved the clarity of the presentation but have also contributed significantly to the overall strength of the research. Your guidance has been instrumental in refining the key aspects of the paper. Please feel free to reach out if further clarification or information is required.

On behalf of all the co-authors

Yours sincerely

Ewa Kostrzeba

Reviewer 2 Report

Comments and Suggestions for Authors

1.Title can be condensed and improved as 'Factors beyond body mass index associated with cardiacm risk among obese children'

2.Authors have taken wide range of age groups (o-19 yrs). Will it be better if the children are analysed in varying age groups?. All body changes may not occur at the same time

3.There are multiple Tables and figures. Can the date be presented in two Tables and two figures?. One for physical and another for biochemical

4.Usually Titles and legends of figures are placed below the figures

Author Response

(The authors gave the same response as above.)

Round 2

Reviewer 1 Report

Comments and Suggestions for Authors

Thank you for thoroughly revising the manuscript based on the comments. I believe the manuscript has improved significantly compared to the original submission. I would recommend accepting the paper at the editor’s discretion.